# miR-205 Regulates the Fusion of Porcine Myoblast by Targeting the Myomaker Gene

**DOI:** 10.3390/cells12081107

**Published:** 2023-04-07

**Authors:** Jideng Ma, Yan Zhu, Xiankun Zhou, Jinwei Zhang, Jing Sun, Zhengjie Li, Long Jin, Keren Long, Lu Lu, Liangpeng Ge

**Affiliations:** 1Chongqing Academy of Animal Sciences, Chongqing 402460, China; 2Key Laboratory of Pig Industry Sciences, Ministry of Agriculture, Chongqing 402460, China; 3Chongqing Key Laboratory of Pig Industry Sciences, Chongqing 402460, China; 4Farm Animal Genetic Resource Exploration and Innovation Key Laboratory of Sichuan Province, Sichuan Agricultural University, Chengdu 611130, China; 5Technical Engineering Center for the Development and Utilization of Medical Animal Resources, Chongqing 402460, China

**Keywords:** pig, Myomaker, myoblast fusion, miR-205, muscle regeneration

## Abstract

Skeletal muscle formation is an extremely important step in animal growth and development. Recent studies have found that TMEM8c (also known as Myomaker, MYMK), a muscle-specific transmembrane protein, can promote myoblast fusion and plays a key role in the normal development of skeletal muscle. However, the effect of Myomaker on porcine (*Sus scrofa*) myoblast fusion and the underlying regulatory mechanisms remain largely unknown. Therefore, in this study, we focused on the role and corresponding regulatory mechanism of the *Myomaker* gene during skeletal muscle development, cell differentiation, and muscle injury repair in pigs. We obtained the entire 3′ UTR sequence of porcine *Myomaker* using the 3′ RACE approach and found that miR-205 inhibited porcine myoblast fusion by targeting the 3′ UTR of *Myomaker*. In addition, based on a constructed porcine acute muscle injury model, we discovered that both the mRNA and protein expression of *Myomaker* were activated in the injured muscle, while miR-205 expression was significantly inhibited during skeletal muscle regeneration. The negative regulatory relationship between miR-205 and Myomaker was further confirmed in vivo. Taken together, the present study reveals that Myomaker plays a role during porcine myoblast fusion and skeletal muscle regeneration and demonstrates that miR-205 inhibits myoblast fusion through targeted regulation of the expression of Myomaker.

## 1. Introduction

Skeletal muscle is a dense striated muscle tissue that plays a key role in regulating body metabolism and homeostasis, accounting for ~40% of body weight. Skeletal muscle is formed by myogenic progenitor cells (MPCs) originating from the multifunctional mesodermal precursor cells [1], labeled by the paired box transcription factors Pax3 and Pax7, which are responsible for skeletal muscle formation [2]. Subsequently, myoblasts undergo proliferation, differentiation, and fusion to form multinucleated myotubes in response to myogenic regulators (MRF, including MyoD, MyoG, Myf5, and MRF4) [3,4]. MPCs produce a subpopulation of cells called muscle satellite cells (MSCs), which contribute to the regeneration of adult muscles [4]. MSCs, also known as adult muscle stem cells, play an important role in mediating the regenerative capacity of adult skeletal muscle, which makes the muscle regeneration model a feasible method to study gene function during skeletal muscle development in vivo. In adult muscles, MSCs are in quiescence. However, once the skeletal muscle is injured, MSCs will be activated immediately to proliferate and differentiate into new muscle fibers, so as to repair the damage [5]. During this process, myoblast fusion is a key step in skeletal muscle formation, but its regulatory network remains to be illustrated.

In 2013, the Millay team discovered that TMEM8c (named Myomaker) is a muscle-specific transmemental protein that directly regulates myoblast fusion [6]. *Myomaker* is highly expressed in developing skeletal muscles and is down-regulated after the completion of muscle formation. In the process of myogenesis and muscle regeneration, the instantaneous expression of Myomaker effectively promotes myoblast fusion [6,7]. In the C2C12 cell line, the expression of *Myomaker* increases sharply during differentiation and fusion, and decreases rapidly at the end of differentiation. Meanwhile, a Western blot analysis showed that the pattern of *Myomaker* protein expression is similar to its mRNA [8], and studies of Myomaker in zebrafish and chickens have reconfirmed these results [9,10]. Nevertheless, the expression pattern and fusion function of Myomaker in pigs have not been studied yet.

MicroRNAs (miRNAs) are a class of non-coding small RNAs with a length of ~22 nt in eukaryotes [11]. They usually bind to the 3′ UTR of target mRNAs, leading to the degradation of mRNAs or inhibition on post-transcriptional translation [12]. Existing evidence suggests that some miRNAs and myogenic regulators play an essential role in myoblast differentiation and fusion processes by regulating the expression of *Myomaker*, which in turn affects myoblast fusion [13,14,15]. A study in mice indicated that miR-491 could specifically bind to the 3′ UTR of *Myomaker*, down-regulate *Myomaker* expression, and inhibit myoblast differentiation [8]. During avian myoblast differentiation, miR-140-3p inhibits *Myomaker* expression and skeletal muscle formation by targeting and binding to the 3′ UTR of *Myomaker* [10]. Another study in geese found four miRNAs whose expression negatively correlated with *Myomaker* expression in goose pectoral and leg muscles, including miR-125b-5p, miR-15a, miR-16-1, and miR-23. Further validations showed that only miR-16-1 could target and bind the *Myomaker* 3′ UTR, suggesting it to be a potential factor that regulates skeletal muscle formation in geese [16]. In summary, miRNAs could serve as critical regulators in the formation of skeletal muscle, and in-depth research on the growth and development process of skeletal muscle demands better knowledge in the regulatory mechanisms of miRNAs in skeletal muscle formation.

Here, we profiled *Myomaker* expression (both mRNA and protein) during pig skeletal muscle development and primary myoblast differentiation and determined the function of *Myomaker* in myoblast fusion by overexpression assay. To further investigate transcriptional regulation mechanisms of *Myomaker*, we amplified the 3′ UTR region of *Myomaker* using 3′ RACE, and found that miR-205 inhibits *Myomaker* expression and pig myoblast fusion by binding to *Myomaker* 3′ UTR in vitro. We also found that the protein expression of *Myomaker* was activated to repair the damage. Taken together, our results demonstrate that Myomaker is critical for pig myoblast fusion, and miR-205 regulates myoblast fusion by targeting the 3′ UTR of *Myomaker*.

## 2. Materials and Methods

### 2.1. Animals and Tissues Collection

The experimental procedures used in this study were approved by the Institutional Animal Care and Use Committee of Sichuan Agricultural University (Approval No. DKY-S20153307, 15 November 2015). A total of thirty-six female Landrace pigs (aged at E85, E113, D0, D180, D270, and Y2, respectively) were used in this study, and there were six biological repeats at each time point. The collected tissues (heart, liver, spleen, lung, kidney, and longissimus dorsi muscles) were immediately frozen in liquid nitrogen, then stored at -80 °C for subsequent experiments. The longissimus dorsi muscles of D0 Landrace were immediately used for primary skeletal muscle satellite cell isolation.

### 2.2. Total RNA Isolation, Reverse Transcription, and qRT-PCR

Total RNAs were isolated from tissues or primary MSCs using RNAiso Plus reagent (TaKaRa, Tokyo, Japan), in accordance with the manufacturer’s protocol. RNA concentration and quality were examined using a NanoDropTM 2000 spectrophotometer (Thermo Fisher Scientific, Woltham, MA, USA). For mRNA, RNA was reverse transcribed to cDNA using the PrimeScriptTM RT reagent kit (TaKaRa, Tokyo, Japan). For MicroRNA, cDNA was synthesized using Mir-XTM miRNA First-Strand Synthesis Kit (TaKaRa, Tokyo, Japan). cDNA was examined by qRT-PCR using the TB GreenTM Premix Ex Taq reagents (TaKaRa, Tokyo, Japan) with specific primers and the CFX Connect Real-Time System (Bio-Rad, Hercules, CA, USA). *GAPDH* and *β-actin* or *U6* were used as housekeeping genes for normalizing mRNA and miRNA expression, respectively. The primer sequences used for the qPCR are listed in Appendix A. All primers used in this study were synthesized by TSINGKE (TSINGKE, Beijing, China).

### 2.3. The 3′ Rapid Amplification of cDNA Ends (3′ RACE) of Myomaker

Total RNA was isolated from E85 Landrace Longissimus dorsi and was reverse transcribed to a first-strand cDNA template using the 3′ RACE Oligo(dT)-anchor primer and PrimeScriptTM II Reverse Transcriptase (TaKaRa, Tokyo, Japan). Next, we amplified the Myomaker 3′ UTR using nested PCR. Specifically, *Myomaker* specific outer primer and the 3′-adaptor outer primer were used for the first round of PCR amplification. We obtained the specific products in the first round; then, the products were gel-purified, ligated into pGEM-T Easy vector (Promega, Madison, WI, USA), and sequenced. All of the primers used in 3′ RACE are listed in Appendix A.

### 2.4. Western Blotting Analysis

Tissue samples of 20–25 mg were taken, and 500μL of RIPA lysis buffer containing 1 mM of PMSF and 0.02% protease phosphatase inhibitors were added for homogenization. Subsequently, the homogeneous liquid was incubated at 4 °C for 30 min. Insoluble substances were removed from the suspension by 12,000× *g* of centrifugation for 15 min, and total protein concentration was quantified by BCA protein assay (Beyotime, Shanghai, China). The protein suspension was electrophoresed in 12% SDS polyacrylamide gels and transferred to polyvinylidene difluoride (PVDF) membranes (Bio-Rad, Hercules, CA, USA) by the Trans-Blot Turbo transfer system (Bio-Rad, Hercules, CA, USA), then blocked with 5% non-fat dry milk in Tris-buffered saline for 2 h at room temperature. Finally, the anti-TMEM8c (dilution 1:500; NOVUS, Littleton, CO, USA) and anti-Tubulin (Abcam, Cambridge, MA, USA) were added separately and incubated overnight at 4 °C (or 2 h at room temperature), followed by 1 h with Alexa Fluor^®^ 488 secondary antibody (Abcam, Cambridge, MA, USA). After incubating the membrane with chemiluminescence reagent (Beyotime, Shanghai, China), the protein expression was detected by a ChemiDoc MP Imaging System (Bio-Rad, Hercules, CA, USA), and anti-Tubulin (Abcam, Cambridge, MA, USA) was used as a housekeeping protein for normalizing Myomaker.

### 2.5. Primary Skeletal Muscle Satellite Cells (MSCs) Isolation, Purification, and Culture

The longissimus dorsi muscle of D0 Landrace was isolated into a Petri dish and washed with PBS (Hyclone, Logan, UT, USA) three times under a sterile environment. Muscle tissue was cut to about 1 mm in length with ophthalmic scissors and digested in a water bath at 37 °C with 0.1% type I collagenase for 1–2 h, which resulted in loose muscle fibers. The MSCs were released after digestion with 0.25% trypsin for 10–20 min [17,18]. Digestion was terminated with high-glucose Dulbecco’s Modified Eagle Medium (DMEM, Hyclone, Logan, UT, USA) supplemented with 20% FBS (GIBCO, Grand Island, NY, USA), 100,000 units/L of penicillin sodium, and 100 mg/L of streptomycin sulfate (Hyclone, Logan, UT, USA). After blowing, 100 μm and 70 μm cell screens were used successively for filtration, and the filtrate was collected and centrifuged for 1000 rmp for 10 min. After the supernatant was discarded, it was suspended again; then, 70% and 40% percoll solution (GE Healthcare, Beijing, China) was used for discontinuous density gradient centrifugation, which was combined with the differential rate adherent method for cell purification [19]. Finally, the MSCs were cultured in DMEM containing 20% FBS at 37 °C in a humidified atmosphere containing 5% CO_2_. After about 24 h, the fluid was changed, and passage could be carried out when the cell density reached 70–80%. Differentiation medium (DM) containing 2% horse serum (GIBCO, Grand Island, NY, USA), 100,000 units/L of penicillin sodium, and 100 mg/L of streptomycin sulfate was used to induce the differentiation of the MSCs.

### 2.6. Immunocytochemistry and Immunohistochemistry

The steps of Immunocytochemistry are as follows: The porcine primary MSCs were washed with PBS 3 times, fixed with 4% paraformaldehyde for 20–30 min, washed with PBS 3 times, then permeated with 0.5% Triton X-100 for 15 min, and finally, blocked with 5% non-fat dry milk in Tris-buffered saline for 1 h at room temperature. Next, these cells were incubated overnight in mouse anti-MyHC (dilution 1:400; Abcam, Cambridge, MA, USA) and anti-Pax7 (dilution 1:250; mouse monoclonal antibody; Abcam, Cambridge, MA, USA) at 4 °C. The next day, the cells were incubated with Alexa Fluor^®^ 488 secondary antibody (Abcam) for 1 h, and the cell nucleus was stained by DAPI (Beyotime, Shanghai, China).

The steps of Immunohistochemistry are as follows: Fixed tissues were dehydrated using a full-automatic dehydrator, paraffin-embedded, and then sectioned into 5 μm thick samples. First, the dewaxed sections were placed into the dyeing tank with 3% methanol hydrogen peroxide at room temperature for 10 min. The samples were rinsed with PBS three times for 5 min each. The slices were dipped into 0.01 M of citrate buffer (pH 6.0) and then heated in the microwave until boiling for 5 min interval; the heating was repeated once more. After cooling, the slice was washed with PBS two times for 5 min each. The sections were then blocked with a blocking serum (ZLI-9021, ZSGB-BIO) at room temperature for 20 min. The sections were incubated with the anti-TMEM8c (1:100; NOVUS Biologicals Littleton, CO, USA) at 4 °C overnight and then with Alexa Fluor^®^ 488 secondary antibody (1:250; Abcam) for 30 min at 37 °C. The samples were rinsed with PBS three times for 5 min each and then processed with the Concentrated DAB kit (K135925C, ZSGBBIO). The sections were then dehydrated in alcohol, cleared in xylene, and mounted in synthetic resin.

The images of the samples were captured using an Olympus IX53 microscope (Olympus, Tokyo, Japan) with cellSens Standard software (v1.16, Olympus Instruments, Tokyo, Japan). Each tissue captures up to six view fields. The mean density of immunohistochemical staining was measured using Image-Pro Plus (IPP).

### 2.7. Overexpressed Plasmid Construction and Transfection

The CDS region of Myomaker was obtained from NCBI, and the sequences were then cloned into the overexpressed vector pcDNA3.1 (Promega, Madison, WI, USA). This recombinant vector was synthesized by TSINGKE (TSINGKE, Beijing, China) and called pcDNA3.1-Myomaker. When MSCs reached 70–80% confluence, Myomaker overexpression plasmids (pcDNA3.1-Myomaker) or empty plasmids (pcDNA3.1-empty) were transfected into the cells using Lipofectamine 3000 (Invitrogen, Grand Island, NY, USA), according to the manufacturer’s instructions. After transfection for 48 h, the MSCs were induced to differentiate.

### 2.8. Dual-Luciferase Reporter Assay

Myomaker sequences that contain miRNA binding sites were cleaved using SacI/XhoI and cloned into the pmirGLO plasmid (Promega, Madison, WI, USA). We named the recombinant pmirGLO vectors with the Myomaker sequence as pmirGLO-Myomaker, which were also synthesized by TSINGKE (TSINGKE, Beijing, China). Meanwhile, the corresponding information of miRNA sequences was found from miRbase. miR-205 mimics (50 nM) and NC mimics (50 nM) were purchased from RIBOBIO (RIBOBIO, Guangzhou, Guangdong, China). When Hela cell density in a 48-well plate reached 70%, pmirGLO-Myomaker was co-transfected with miRNA mimic into Hela cells using Lipofectamine 3000, according to the manufacturer’s instructions. Cells were collected after 48 h; dual-luciferase activity was measured using the Dual-Luciferase Reporter Assay System kit (Promega, Madison, WI, USA), according to the manufacturer’s instructions.

### 2.9. In Vivo Muscle Injury

Bupivacaine is considered one of the most toxic local anesthetics, which can cause skeletal muscle damage and induce skeletal muscle repair [20]. Here, 0.5% bupivacaine hydrochloride monohydrate (Fluka, Milwaukee, WI, USA) in physiological saline (0.9% NaCl) was injected into the left longissimus dorsi muscle of a one-month-old Landrace, using sterile syringes, with the injection site the longissimus dorsi muscle between the third and sixth ribs. The needle was inserted parallel to the muscle fiber longitude and then slowly withdrawn while simultaneously injecting the bupivacaine solution in its path. The right longissimus dorsi muscle was used as a control by injecting the same dosage of 0.9% NaCl. The pigs were killed for the collection of target longissimus dorsi muscles at 0 h (non-injected pig), 2 h, and on Days 1, 3, 5, 7. Samples were then collected for qRT-PCR and immunohistochemistry.

### 2.10. Statistical Analysis

Statistical significance was calculated by the Student’s *t*-test for comparisons of two groups or one-way analysis of variance (ANOVA) with Tukey’s post hoc test for multiple groups.

## 3. Results

### 3.1. Amino Acid Evolution Tree and Gene Structure of Myomaker

To construct the phylogenetic relationship of Myomaker among vertebrates, we used MEGA7.0 [21] to compare amino acid sequences and found that mammals are distantly related to fish or birds, in consistency with the tree of life. *Sus scrofa* and *Bos taurus* in mammals are closely related, while *Pan paniscus* and *Homo sapiens*, both primates, have nearly identical amino acid sequences (Figure 1A). To further study the structure of porcine *Myomaker*, we obtained the predicted porcine *Myomaker* sequence from the NCBI database, which is located on Chromosome 1 and contains five exons and four introns. Using the predicted sequence information, we designed a gene-specific primer sequence and amplified the 3′ UTR of porcine *Myomaker* using the 3′ RACE approach for the subsequent functional mechanism analysis (Figure 1B).

### 3.2. Myomaker Expression Pattern during Pig Skeletal Muscle Development

In order to determine whether Myomaker is involved in the fusion of pig myoblasts, we first profiled the expression of Myomaker during pig skeletal muscle development and primary myoblast fusion. The quantitative real-time PCR (qRT-PCR) of Myomaker mRNA in the representative tissues of porcine embryos for 85 days (E85) showed that Myomaker was specifically expressed in skeletal muscle relative to other tissues (one-way analysis of variance (ANOVA), *p* < 0.01; Figure 2A). On the other hand, the mRNA expression of Myomaker was the highest at E85, and decreased with the increase in age (ANOVA, *p* < 0.01; Figure 2B). These results are similar to a previous study in mice [6]. In addition, Western blot showed that the Myomaker protein levels were consistent with Myomaker mRNA expression in different tissues and at different stages (ANOVA, *p* < 0.01; Figure 2C–E).

To investigate the patterns of Myomaker expression in vitro, we first isolated and purified primary muscle satellite cells (MSCs) from pig longissimus dorsi muscle, and immediately identified MSCs (GM) with an anti-Pax7 antibody. Immunofluorescence data showed that the proportion of Pax7+ positive cells was as high as 90% (Figure 2G). Pax7 is a transcriptional factor localized specifically in the nuclei. However, a small part of cytoplasmic was stained by Pax7 (Figure 2G). This may be due to the lack of commercially specific antibodies for the porcine Pax7 gene. Mouse monoclonal antibody was used in this assay. Moreover, the qRT-PCR of Pax7 mRNA showed that Pax7 was highly expressed in MSCs but significantly decreased in porcine intestinal epithelial cells (IPEC) and kidney cells (PK15) (ANOVA, *p* < 0.01; Figure 2H), indicating success in isolating and purifying MSCs from porcine longissimus dorsi. Next, isolated MSCs were cultured and induced to differentiate (Figure 2I). During the process of MSCs differentiation and fusion, the mRNA expression of the Myomaker gene gradually increased and reached the highest level on the fourth day of differentiation (DM4), after which the expression significantly decreased (ANOVA, *p* < 0.05; Figure 2F). In brief, these results indicate tissue-specific expression of Myomaker in skeletal muscle and high expression in embryonic pigs, as well as a peak of Myomaker expression at myoblast fusion.

### 3.3. Myomaker Plays an Important Role in Porcine Primary Myoblast Fusion

Previous studies have shown that overexpression of Myomaker significantly promotes myoblasts fusion, but it is not clear whether this effect is still conserved in pigs. Therefore, we first constructed a Myomaker overexpression vector (pcDNA3.1-Myomaker) to study the effect of Myomaker overexpression on pig myoblast fusion. RT-PCR of Myomaker showed that after transfection with pcDNA3.1-Myomaker for 48 h, the expression level of Myomaker mRNA was about four times that of the control group (Unpaired two-tailed Student’s *t*-test, *p* < 0.01; Figure 3A), indicating that the transfection was successful. Immunostaining with myosin heavy chain (myHC) showed that compared to the control group, overexpression of Myomaker resulted in the formation of large myotubes with more nuclei. The cell fusion index after transfection with Myomaker overexpression vector was also significantly increased (Unpaired two-tailed Student’s *t*-test, *p* < 0.05; Figure 3B,C), suggesting that Myomaker overexpression could significantly promote myoblast fusion. Myoblast fusion is a complex process involving many myogenic regulatory factors, such as MyoD, Myf5, and MyoG. The expression pattern of MyoG is consistent with that of Myomaker. RT-PCR of MyoG, Myf5, and MyoD showed that the expression of MyoG was decreased after transfection with pcDNA3.1-Myomaker (Unpaired two-tailed Student’s *t*-test, *p* < 0.05), while there was no difference in the expression of Myf5 or MyoD (Figure 3D–F). In a word, these results indicated that Myomaker promotes fusions of porcine primary myoblasts.

### 3.4. miR-205 Regulates Myomaker Expression in Pigs

Based on the 3′ UTR sequence of the pig Myomaker that has been amplified by 3′ RACE, we predicted potential target miRNAs using TargetScan [22]. Among these miRNAs, we found the expression levels of miR-205, miR-30b-3p, miR-30c-3p, and miR-491 negatively correlated with that of Myomaker during differentiation of MSCs (Appendix A, Figure 4A). To further determine whether the four miRNAs directly target 3′ UTR of Myomaker mRNA, binding sites between miRNAs and Myomaker, as well as minimal free energy (mfe) were predicted by RNAhybrid [23] (Appendix A). We subsequently performed a dual-luciferase reporter assay in Hela cells to confirm the physical relationship between miRNAs and Myomaker. The result indicated that miR-205-binding sites in Myomaker 3′ UTR were conserved among several representative species (Figure 4B). Next, Myomaker 3′ UTR that contains the miR-205, miR-30b-3p, miR-30c-3p or miR-491 binding site was inserted into dual luciferase plasmid, respectively (pmirGLO-Myomaker 3′ UTR) (Figure 4B). It turned out that miR-30b-3p increased the luciferase activity after transfection with the Myomaker 3′ UTR reporter, while miR-30c-3p and miR-491 had no difference in the luciferase activity. Only miR-205 reduced the luciferase activity after transfection (Unpaired two-tailed Student’s *t*-test, *p* < 0.05), which indicates that Myomaker is a direct target of miR-205 (Figure 4C). A standard validation report for miR-205-Myomaker interaction in this study is shown in Appendix A [24,25].

### 3.5. miR-205 Inhibits Primary Myoblasts Fusion through Targeting Myomaker

To identify the role of miR-205 in myoblasts fusion, miR-205 mimics or negative control (NC) were transfected into porcine primary myoblasts at DM2. RT-PCR of miR-205 showed that after transfection with miR-205 mimics for 48 h, the expression level of miR-205 significantly increased in comparison with NC (Unpaired two-tailed Student’s *t*-test, *p* < 0.01; Figure 5A), indicating that the transfection was successful. The multi-nucleated myotubes were immunostained using anti-myHC, and found that primary myoblasts transfected with miR-205 mimics were incapable of fusing to form multi-nucleated myotubes, in contrast to cells transfected with NC (Figure 5C). Moreover, after transfected with miR-205 mimics, the expression of Myomaker was inhibited and the fusion index of myotube was significantly decreased (Unpaired two-tailed Student’s *t*-test, *p* < 0.01 and *p* < 0.05; respectively Figure 5B,D). We found that miR-205 also inhibited the expression of MyoG (Unpaired two-tailed Student’s *t*-test, *p* < 0.01), while we observed no significant effect on MyoD and Myf5, indicating that miR-205 might have effects at the late differentiation (Figure 5E). Skeletal muscle is composed of different types of myofibers. Myofibers were classified into four types according to their contractile characteristics, energy metabolism differences, oxidation, and fermentation capacity, including myofibers of MyHC1, MyHC2a, MyHC2b, and MyHC2x [26,27,28]. Given that miRNAs play an important regulatory role in the transformation of myofiber types, we investigated the effect of miR-205 on muscle fiber typing. Overexpression of miR-205 reduced the expression of all types of muscle fibers, among which the expression of MyHC2x was significantly reduced (Unpaired two-tailed Student’s *t*-test, *p* < 0.01), suggesting that the effect of miR-205 on myofiber formation may be partly due to its inhibition on MyHC2x expression (Figure 5F).

### 3.6. Expression Profiles of Myomaker during Porcine Muscle Regeneration

To investigate the in vivo effect of Myomaker and miR-205 on muscle regeneration, we constructed a skeletal muscle acute injury model of pigs using bupivacaine (BPVC) (Milwaukee, USA). The pig longissimus dorsi muscle was injured with 0.5% bupivacaine and harvested at 0 h, 2 h, 1, 3, 5, 7 days after injection (Figure 6A). Hematoxylin and eosin (H&E) staining displayed the process of muscle injury and regeneration (Figure 6B). We observed that muscle damage started 2 h after the injection, followed by extensive muscle damage and severe inflammatory infiltration 1 day later (Figure 6B). Muscle repair began at 3 d after injury, and most of the damaged muscles regenerated at 7 d (Figure 6B). The cross-sectional area (CSA) of new muscle fibers gradually increased during the process of muscle regeneration (ANOVA, *p* < 0.01; Figure 6C). In addition, the expression level of Myomaker increased in the muscle regeneration and peaked at 5 d (ANOVA, *p* < 0.01; Figure 6D). The expression of miR-205 was negatively correlated with the expression of Myomaker, which is consistent with the previous results (Figure 6E). Immunohistochemistry with Myomaker showed that the integrated optical density (IOD) of Myomaker protein was similar to its mRNA expression trend (Figure 6F,G). Moreover, we found that the expression levels of MyoD, MyoG, and Myf5 increased gradually during muscle regeneration, reaching a peak at 5 d (ANOVA, *p* < 0.01; Figure 6H–J). These results suggest that once a skeletal muscle becomes damaged, the expression of Myomaker is activated to repair the damage and promote the formation of new muscle fibers.

## 4. Discussion

Before 2013, research on the TMEM8c gene was almost non-existent, until the Millay team [6] found that TMEM8c is a muscle-specific transmembrane protein that directly regulates myoblast fusion, and named it Myomaker. The team found that Myomaker is specifically expressed in muscle tissue, suggesting the importance of Myomaker for skeletal muscle development. Meanwhile, Zhang et al. [9] and Landemaine et al. [29] found that Myomaker is highly expressed in the fast muscles of zebrafish (with strong fusion ability), but significantly decreased in slow muscles (with weak fusion ability). In addition, it was also found that the expression level of *Myomaker* in skeletal muscle formation in embryos is significantly higher than that after skeletal muscle development. Although the expression pattern of *Myomaker* is similar in mice, zebrafish, and chickens [9,10,29], this conservation has yet to be determined in more species. In the present study, we profiled the expression of Myomaker in pigs and found that it was specifically expressed in skeletal muscle tissue. The expression level in embryos was significantly higher than that after birth, which is consistent with the results from the Millay team [6]. In addition, through primary cell culture, we found that the expression level of Myomaker in the fusion process of myoblast cells was significantly higher than that in other periods. These results reconfirmed previous findings [10]. However, myoG expression was inhibited after transfection with pcDNA3.1-Myomaker, which we speculate may be caused by negative feedback. In a word, both in vivo and in vitro experiments demonstrate that *Myomaker* is highly expressed during the development of skeletal muscle in pigs, suggesting its vital role in this biological process.

Bupivacaine is considered one of the most toxic drugs in the skeletal muscle [30,31,32], and the Burn team first discovered that local anesthetics cause changes in muscle histomorphology [33]. Gergin et al. demonstrated that 0.5% bupivacaine causes the most serious damage to muscle tissue [34]. Later, Benoit et al. found that 0.5% bupivacaine can cause skeletal muscle injury and induce skeletal muscle repair [35]. In this article, we successfully constructed a skeletal muscle injury model of pigs by injecting 0.5% bupivacaine, and found that in adult pig skeletal muscles, Myomaker was almost not expressed. However, when the muscles were damaged, *Myomaker* expression was activated to repair the damage. At the same time, the expression patterns of genes that encode myogenic regulatory factors, such as MyoG, MyoD, and Myf5, were similar to those of Myomaker, and their correlations with *Myomaker* expression levels were up to 0.91, 0.81, and 0.76, respectively (Appendix A).

Previous studies have shown that Myomaker and Myomerger, which are both involved in regulating myoblast fusion, are mainly regulated by MRFs and some non-coding RNAs. Luo et al. found that during the differentiation of myoblasts in birds, MyoD and MyoG can bind directly to the Myomaker promoter to regulate Myomaker transcription [10]. MiRNAs also play a crucial role in skeletal muscle differentiation [13,14,15]. For example, miR-1 promotes myoblast differentiation in skeletal muscle [36]. miR-206 is highly expressed in skeletal muscle [37], although its function remains to be studied. Luo, He, and Ke et al. showed that miR-140-3p, miR-491, and miR-16-1, respectively, target the 3′ UTR of *Myomaker* to inhibit the gene’s expression and fusion function [8,10,16]. In this research, we found that miR-205 can also bind to the 3′ UTR of *Myomaker* in pigs, and this binding site is highly conserved among some common mammals. MiR-205 can widely participate in cancer development and was once defined as esophageal squamous cell carcinoma-specific miRNA [38,39]. However, in the current study, we found that the overexpression of miR-205 significantly inhibited the expression and fusion function of Myomaker. In addition, the expression of *MyoG* was also inhibited, suggesting that MyoG may also be involved in myoblast fusion. We also found that miR-205 inhibited the expression of *MyHC2x* in different types of muscle fibers, suggesting that miR-205 also had a certain effect on muscle fiber typing.

There are still some deficiencies in the study: (1) When the primary cells were immunostained, no specific Pax7 antibody of pigs was purchased, and the Pax7 antibody of other mammals was selected. Due to the species specificity, some degree of cytoplasmic nonspecific staining was present. (2) There is no commercial cell line of porcine microsatellite cells. The primary porcine microsatellite cells isolated in this experiment are not as capable of proliferation and differentiation as commercial cell lines, such as mouse C2C12 cell lines. Therefore, the ratio of myoblasts formed by primary culture and the differentiation of pig muscle cells is low, along with the fusion rate of cells. Taken together, further confirmatory studies are encouraged with the more specific cell lines and antibodies.

## 5. Conclusions

Our results indicate that *Myomaker* is highly expressed in the early skeletal muscle development of pigs and primary myoblasts differentiation. Myomaker overexpression promotes the myoblast fusion of pigs. We also demonstrated that miR-205 can inhibit *Myomaker* expression and myoblast fusion in pigs by targeting the 3′ UTR of the gene. In addition, the present study suggests that *Myomaker* expression in injured adult muscle is activated to repair the damage (Figure 7).

## Figures and Tables

**Figure 1 cells-12-01107-f001:**
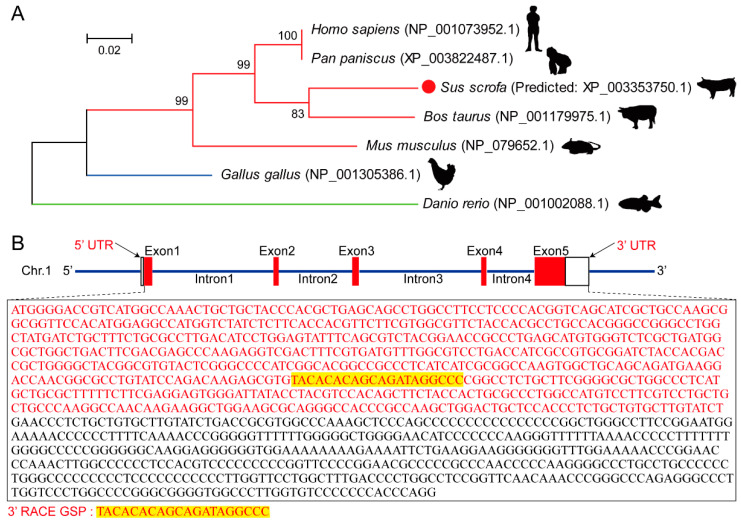
Amino acid evolution tree and gene structure of porcine Myomaker. (**A**) Evolutionary relationship of Myomaker between *Bos taurus*, *Sus scrofa*, *Homo sapiens*, *Pan paniscus*, *Mus musculus*, *Gallus gallus*, and *Danio rerio*. The values in the figure are Bootstrap values, reflecting the reliability among adjacent branches, and the closer the values are to 100, the higher the reliability; (**B**) gene structure of pig Myomaker. Nucleotides in red represent the CDS region, where 3′ RACE GSP represents the 3′ RACE gene-specific primer (GSP). Nucleotides in black represent the 3′ UTR of *Myomaker* by 3′ RACE.

**Figure 2 cells-12-01107-f002:**
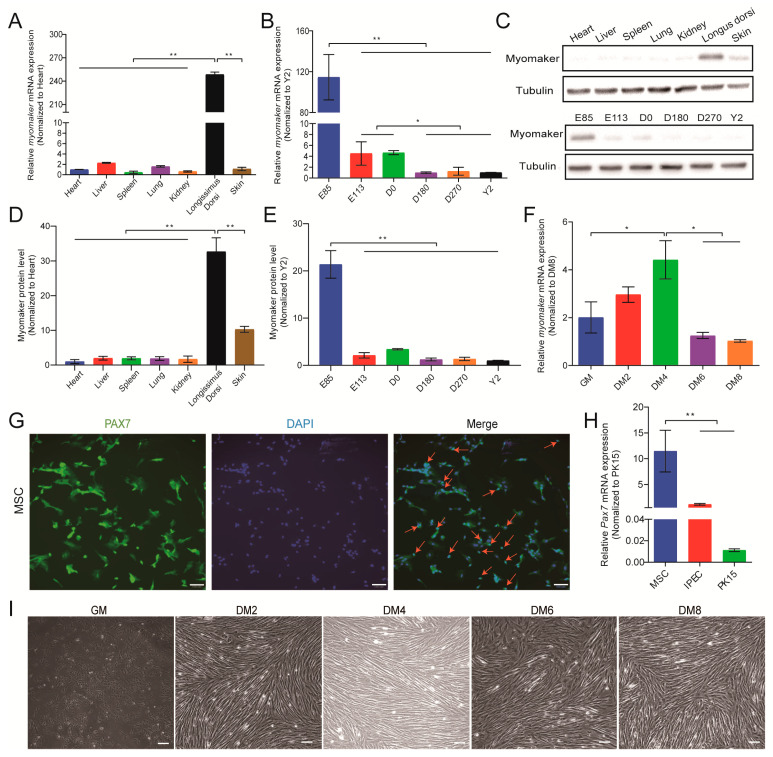
Patterns of Myomaker expression during pig skeletal muscle development. (**A**) Relative mRNA expression of Myomaker in representative tissues (Heart, Liver, Spleen, Lung, Kidney, Longissimus dorsi and Skin) of E85 pig, β-actin and GAPDH were used as reference genes; (**B**) Relative mRNA expression levels of Myomaker in pigs at different ages (E85: Embryonic Day 85; E113: Embryonic Day 113; D0: Day 0 of life; D180: Day 180 of life; D270: Day 180 of life; Y2: year 2 of life); (**C**–**E**) Protein levels of Myomaker at different tissues and ages were detected by Western blot and normalized to Tublin. (**F**) Relative mRNA expression levels of Myomaker during proliferation (GM: Growth medium, 50%) and differentiation (DM2: Differentiation culture for 2 days; DM4: Differetiation culture for 4 days; DM6: Differentiation culture for 6 days; DM8: Differentiation culture for 8 days) of porcine MSCs. (**G**) Immunofluorescence with an anti-Pax7 antibody. Red arrow, Pax7 specific positive cells in nucleus. Bar, 100 µm; (**H**) Relative mRNA expression levels of Pax7 in MSCs, IPEC, and PK15; (**I**) Phase-contrast micrographs of proliferation (GM, 50%) and differentiation (DM2, DM4, DM6, and DM8) of pig MSCs. Bar, 100 µm; Values in A and B are mean ± S.D. with six pigs per group. Values E and G are mean ± S.D. with three cultures per group. ns = not significant; * *p* < 0.05; ** *p* < 0.01 compared between corresponding groups.

**Figure 3 cells-12-01107-f003:**
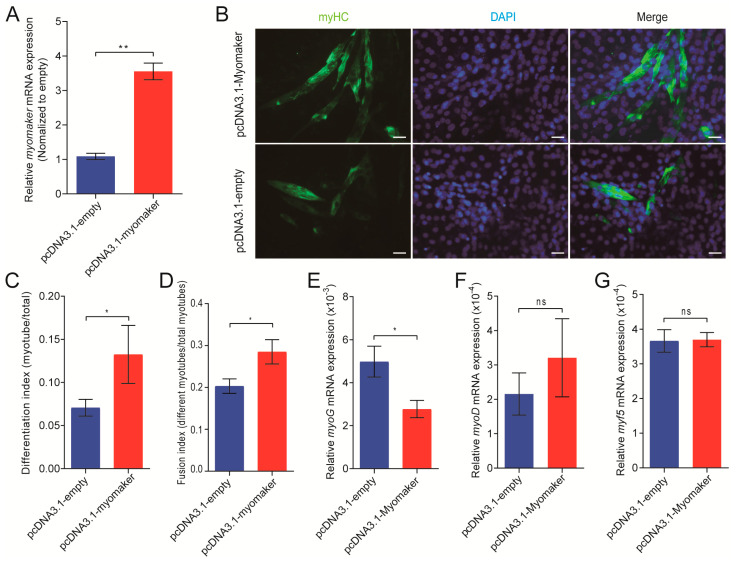
Myomaker plays an important role in pig myoblasts fusion. (**A**) Relative mRNA expression levels of Myomaker after transfection with pcDNA3.1-Myomaker or pcDNA3.1 empty vector at 48 h; (**B**) Immunostaining with anti-MyHC antibody was performed on MSCs after transfected with pcDNA3.1-Myomaker or pcDNA3.1-empty vector at 48 h. Bar, 50 µm; (**C**,**D**) The differentiation index and fusion index of MSCs after transfection was calculated; (**E**–**G**) Relative mRNA expression levels of MyoG, MyoD, and Myf5 after transfection. These data are expressed as mean ± S.D. of three independent experiments. ns = not significant; * *p* < 0.05; ** *p* < 0.01 compared with negative control.

**Figure 4 cells-12-01107-f004:**
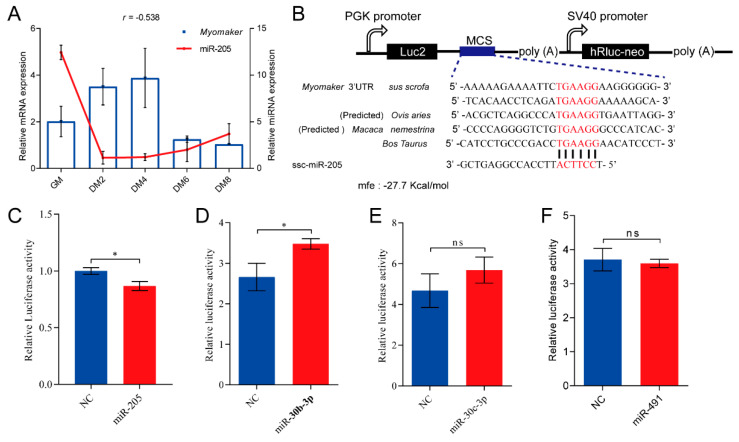
miR-205 regulates Myomaker expression in pigs. (**A**) Correlation between Myomaker and miR-205 expression during primary myoblast differentiation (GM: Growth medium; DM2: Differentiation culture for 2 days; DM4: Differetiation culture for 4 days; DM6: Differentiation culture for 6 days; DM8: Differentiation culture for 8 days); (**B**) Conservation of miR-205-binding site in Myomaker 3′ UTR among several representative species (mfe: minimal free energy); (**C**) Luciferase reporter assay indicated that transfection with miR-205 mimic significantly suppressed the relative activity of luciferase. (**D**–**F**) Relative luciferase activity of miR-30b-3p, miR-30c-3p, and miR-491 after transfection.These data are represented as mean ± S.D. of three independent experiments. ns = not significant; * *p* < 0.05 compared to negative control.

**Figure 5 cells-12-01107-f005:**
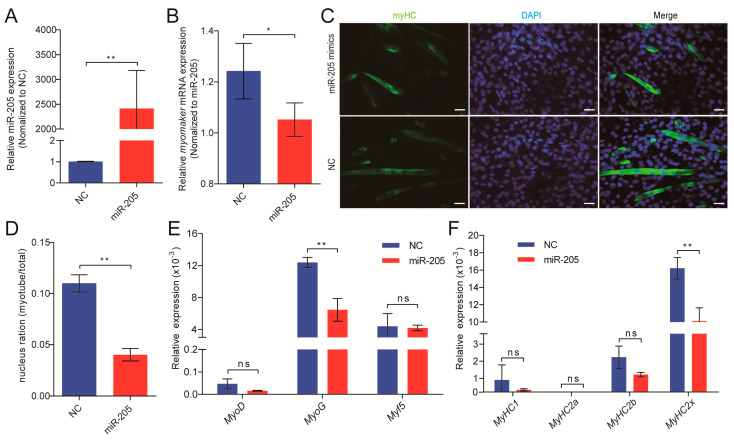
miR-205 inhibits primary myoblasts fusion and Myomaker expression. (**A**) The relative expression levels of miR-205 after transfection with miR-205 mimics or NC for 48 h; (**B**) the relative expression levels of Myomaker after transfection with miR-205 mimics or NC; (**C**) immunostaining with anti-MyHC antibody was performed on primary myoblasts after transfection with miR-205 mimics or NC. Bar, 50 µm; (**D**) the fusion index of primary myoblasts after transfection was calculated; (**E**) the relative expression levels of marker genes of skeletal muscle differentiation (early phase: MyoD and Myf5; later phase: MyoG) after transfection; (**F**) the relative expression levels of MyHC1, MyHC2a, MyHC2b, and MyHC2x after transfection with miR-205 mimics or NC. Results are represented as mean ± S.D. of three independent experiments. ns = not significant; * *p* < 0.05; ** *p* < 0.01 compared to negative control or between the indicated groups.

**Figure 6 cells-12-01107-f006:**
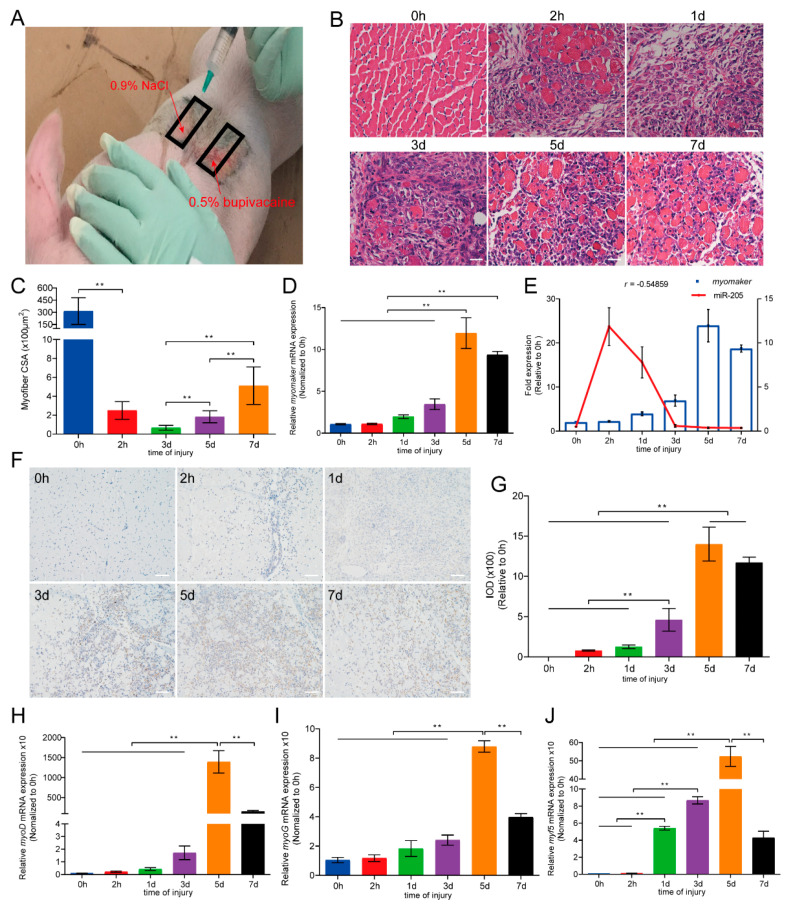
Expression profiles of Myomaker during muscle regeneration. (**A**) Skeletal muscle acute injury model of pig longissimus dorsi muscle by injecting 0.5% bupivacaine; (**B**,**C**) H&E staining of longissimus dorsi muscle and the quantification of muscle fiber CSA. Bars, 20 µm; (**D**,**E**) the relative expression levels of Myomaker mRNA was examined by quantitative real-time PCR and correlation between expression of Myomaker and miR-205; (**F**,**G**) immunohistochemistry with Myomaker and the quantification of IOD. Bars, 50 µm; (**H**–**J**) The relative expression levels of marker genes of skeletal muscle differentiation (early phase: MyoD and Myf5, later phase: MyoG). Results are represented as mean ± S.D. of three independent experiments. ** *p* < 0.01 compared between the indicated groups.

**Figure 7 cells-12-01107-f007:**
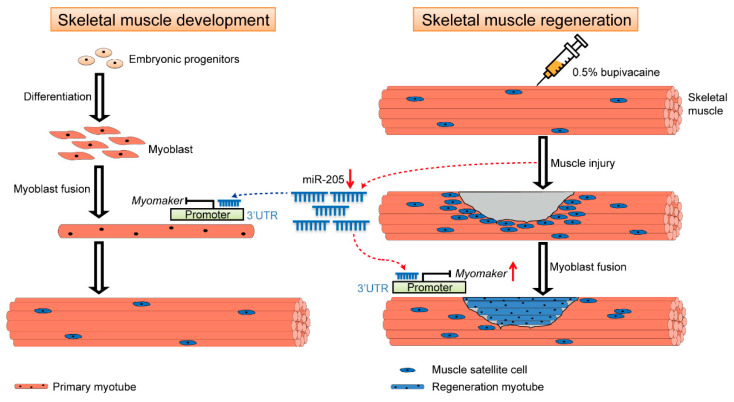
Schematic showing the regulation pattern of miR-205 on skeletal muscle. MiR-205 can regulate *Myomaker* expression by binding to the *Myomaker* 3′ UTR during porcine skeletal muscle development, which in turn affects myoblast fusion to form primary myotubes. After bupivacaine-induced porcine skeletal muscle injury, the expression of miR-205 was down-regulated, which attenuated the inhibitory effect on *Myomaker* expression and up-regulated the expression of *Myomaker*, thereby promoting the fusion of muscle satellite cells into regeneration myotube, which in turn mediated the repair of bupivacaine-induced porcine skeletal muscle injury.

## Data Availability

Not applicable.

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
