# Peer review of "miR-205 Regulates the Fusion of Porcine Myoblast by Targeting the Myomaker Gene"

_cells, 2023, doi:10.3390/cells12081107_

Round 1

Reviewer 1 Report

Myoblast fusion is a key biological process in skeletal muscle development. Exploring the regulatory mechanisms of myoblast fusion is helpful to broaden the knowledge of muscle formation and the pathogenesis of muscle diseases. In this study, the authors reported that Myomaker is highly expressed in early skeletal muscle development of pigs and primary myoblast differentiation. Myomaker overexpression promotes myoblast fusion in pigs. They also demonstrated that miR-205 can inhibit Myomaker expression and myoblast fusion in pigs by targeting the 3’ UTR of Myomaker. The Results from this study will lead to mechanistic insights into the skeletal muscle development, especially myoblast differentiation and fusion processes in pigs.

Major Comments:

1.        Only the transcriptional activity of myomaker after overexpressing miR-205 is not enough to prove the direct regulatory relationship between them. After mutation of miR-205 binding site, it is necessary to detect its binding activity with the sequence before and after mutation.

2.        Why did you choose E85 as your experimental time point? What are the characteristics of muscle development at this stage compared with others?

3.        Figure 3B, C: Specifically, myoblast differentiation and fusion are two biological events during skeletal muscle formation. I think authors should statistically analyze differentiation index and fusion index separately, with the fusion index represents the percentage of nuclei in fused myotubes (containing two or more nuclei) out of the total nuclei, and the fusion index represents the percentage of nuclei in all MyHC positive cells out of the total nuclei.  

Mini comments:

1. Lines 32-33: “reveal” and “demonstrate” should be “reveals” and “demonstrates”, respectively.

2. Line 34: “through targeted regulation of Myomaker expression” should be replaced with “through targetedly regulating the expression of Myomaker”.

3. Lines 40-42: The description is wrong. Myogenic progenitor cells (MPCs), originated from the multifunctional mesodermal precursor cells, labeled by the paired box transcription factors Pax3 and Pax7, are responsible for skeletal muscle formation.

4. Line 49: "injury" is useless in this sentence.

5. "in vivo" and " in vitro" do not have the same format. Some of them are not italics.

6. Lines 119-120: The punctuation is wrong and there are grammatical errors.

7. Line 125: 500L or 500μl ?

8. Line 129: Authors should provide commodity information for the BCA protein concentration kit.

9. Line 203: I suggest replacing "the same amount" with "the same dosage".

10. Line 208: "5 mm thick sample" could be wrong.  5μm thick sections?

11. Line 255: The "decreases" should be changed to the past tense “decreased”.

12. Line 258: I suggest replacing "at different ages" with the more accurate "at different stages", because it is not appropriate to describe the embryonic period with age.

13. Line 299: “Myomaker” should be replaced with “Myomaker overexpression vector”.

14. Figure 2C: There is a typing error in “tublin”. Tubulin?

15. There is no legend for Fig4D, E, F.

16. Lines 332-333: “which indicates that miR-205 is a direct target for Myomaker” should be revised to “which indicates that Myomaker is a direct target of miR205”.

17. Line 452: I suggest replacing “paper” with “research”.

Reviewer 2 Report

In this manuscript the authors focus on the regulatory mechanism of Myomaker gene during skeletal muscle development, cell differentiation, and muscle injury repair in pig. The negative regulatory relationship between miR‐205 and Myomaker was further confirmed in vivo. All in all, the present study reveal that Myomaker plays a role during porcine myoblast fusion and skeletal muscle regeneration, and demonstrate that miR‐205 inhibits myoblast fusion through targeted regulation of Myomaker expression. This study offer useful information for deep understanding the muscle development .

Major points

1. The authors first construct the phylogenetic relationship of Myomaker among vertebrates, but did not further explain this relationship. For example, what kind of differences in Myomaker specifically exist between species?

2. In the author's description, mRNA expression of Myomaker was the highest at E85, and decreases with the increase of age. If possible, early stages chould be studied here to demonstrate that Myomaker expression peaks at this time point.

3. MyoG expression decreased after overexpression of Myomaker, this result seems to be inconsistent with the existing studies. 

Minor points

1.Line 50: “In adult muscles, MSCs are in quiescence.” References should be added here.

2. Antibodies and other reagents should note the item number.

3. The sentence should start with a capital letter.

Line 119: specifically, using Myomaker specific outer primer and the 3’‐adaptor outer primer for the first round of PCR amplification.

4. Gene name should be indicated in italic type.

Line 42-43: These precursor cells further determine their development into myoblasts by expressing Pax3 and Pax7.

Line 83-85: Here, we profiled Myomaker expression (both mRNA and protein) during pig skeletal muscle development and primary myoblast differentiation, and determined the function of Myomaker in myoblast fusion by overexpression assay. 

5. Gene name should be written uniformly in manuscript.

Line 43-45: Subsequently, myoblasts undergo proliferation, differentiation, fusion to form multinucleated myotubes in response to myogenic regulators (MRF, including MyoD, MyoG, Myf5, and MRF4) 

Line 356-359: We found that miR‐205 also inhibited the expression of myoG (Unpaired two‐tailed Student’s t‐test, p < 0.01), while we observed no significant effect on myoD and myf5, indicating that miR‐205 might have effects at the late differentiation (Figure 5E). 

Reviewer 3 Report

In this study, the investigators analyzed the expression and function of TMEM8c (also known as Myomaker) in porcine (Sus scrofa) myoblast fusion and the regulation of MYMK expression by miR-205.  They demonstrated that MYMK expression increased dramatically during muscle cell differentiation in vitro and muscle regeneration in vivo.  Data of these studies are consistent with previous findings from mice and other vertebrate animals.  The identification of miR-205 in repression of MYMK expression is novel. However, all the functional assays were performed using primary cells isolated from skeletal muscles. They investigators believed that these cells are skeletal muscle satellite cells (MSCs), and performed expression, knockdown and dual luciferase reporter assays using these cells to show that miR-205 targets to MYMK3’-UTR and downregulates MYMK expression. Several major concerns are associated with the claimed skeletal muscle satellite cells (MSCs). 

1. In Fig. 2G, the investigators carried out PAX7 staining on primary cells isolated from skeletal muscles.  It is not clear when the immunostaining was performed.  Was it performed on cells in GM or DM ? Pax7 is transcriptional factor localized specifically in the nuclei.  Fig. 2G showed both nuclear and cytoplasmic staining, raising the concern with specificity.  In addition, Pax 7 is only expressed in satellite cells (muscle system cells).  It is surprising to see most of the cell in Fig. 2G were pax7 positive, again raising a concern with specificity.     

2. In Fig. 2I, the investigators cultured the cells in differentiation medium for 2, 4, 6 and 8 days.  If the primary cells are satellite cells or myoblasts, they should differentiate into multinucleated fibers.  Were the cells in DM2-DM8 multinucleated fibers? It appeared as fibroblasts. Immunostaining with anti-myosin antibody should be performed and high magnification imaging should be taken to demonstrate that the so called MSCs were true muscle stem cells that were able to differentiate into myofibers.

3. In fig. 3, the investigators used MYMK overexpression in MSCs to demonstrate its function in inducing myoblast fusion.   MYMK is normally expressed endogenously in muscle stem cells and myoblasts and can drive myoblast fusion without ectopic expression of MYMK.  The use of primary muscle cell culture system was not appropriate for this assay.  The poor fusion efficiency again questions the nature of the primary cells.

4. In Fig. 5C, a few multinucleated fibers were clearly detected ted in the NC. The efficiency was however very poor, suggesting that may be only a small percentage of these primary cells were muscle stem cells or myoblasts, thus questioning the suitability of using these cells for the functional assay. 

5. In fig. 6E, miR-205 and MYMK showed the opposite profiles of expression during muscle regeneration. The RT-PCR data could not reveal if the opposite pattern of expression occurred in the same cell types or due to changes of cell type population during muscle regeneration.  In normal healthy muscle (before muscle damage), MYMK expression was very low. Interestingly miR-205 expression was also very low.  If miR205 was involved in MYMK repression, could you expect to detect a higher level of miR-205 expression?

6. Fig. 1.  3’RACE was used to isolated additional 3’-UTR sequence from pig MYMK.   Was the 3’UTR complete? If it was complete, should it be possible to see part of the polyA sequence?

Reviewer 4 Report

This manuscript systematically studied the negative regulatory relationship between miR-205 and Myomaker at the levels of pig skeletal muscles, MSCs and skeletal muscle acute injury model, and also the results could well clarify the regulatory function and molecular mechanism of miR-205 and Myomaker in the fusion of porcine myoblast. However, some points need to be improved.

1.     Normally miRNA mimics and inhibitor are both used to study the inhibition effects of miRNA and its target from the overexpression and inhibition level. But in this study just miR-205 mimics was used to investigate the effects of miR-205 on Myomaker expression and primary myoblasts fusion. So what’s the results would be if you try to use inhibitor?

2.     In the section of 2.4 Western blotting Analysis on L125, “Tissue samples of 20-25 mg were taken, and 500 L RIPA lysis buffer…… were added ” should be checked for correction.

3.     In the section of Materials and Methods, “Immunocytochemistry analysis” of 2.6 and 2.10 should be combined and reorganized.

4.     In Dual-luciferase reporter assay, usually a mutated vector for the target sequence also should be designed. But this article just used a wide-type vector of target gene.

5.     Gene names when they mean the expression on mRNA level should be italicized in the full text.

6.     The manuscript is to explore the roles and regulatory mechanism of miR-205 and its target Myomaker in porcine muscle fusion, but it was too little about the role of miR-205 in the discussion. So miR-205 function should be highlighted based on its expression etc..

Round 2

Reviewer 1 Report

My concerns have been basically addressed in the revised manuscript. I think it could be accepted in present form.

Reviewer 2 Report

All the questions have been revised. I don't have any more questions. 

Reviewer 3 Report

Most of my previous comments have not been addressed, such as the Pax7 antibody specificity, the characterization of myoblast differentiation using immunostaining with anti-myosin antibody, and the poor fusion efficiency.  I am concerned with the nature of primary cells used in the various assays, and thus may impact the conclusion of this study.

Reviewer 4 Report

I basically agree the authors’ reply and the revised version, except for the action for the suggestion “In the section of Materials and Methods, “Immunocytochemistry analysis” of 2.6 and 2.10 should be combined and reorganized.” Authors just simply combined these two parts but not reorganized. Otherwise, authors can use the original format of this section.

Round 3

Reviewer 3 Report

1. The lack of anti pig-specific Pax7 antibody is not a good explanation.  There is a good anti-pax7 antibody that has been successfully used in many animal species.   https://dshb.biology.uiowa.edu/PAX7

2. Immunostaining of primary cultured cells using anti-myosin antibody is critical to get a sense of the percentage of primary cultured cells are myoblasts/myocytes.

Reviewer 4 Report

Authors have been modified the manuscript according to the suggestion.